# Biodegradation of Nitrile Gloves as Sole Carbon Source of *Pseudomonas aeruginosa* in Liquid Culture

**DOI:** 10.3390/polym16081162

**Published:** 2024-04-20

**Authors:** Javier Alejandro Delgado-Nungaray, David Grajeda-Arias, Eire Reynaga-Delgado, Orfil Gonzalez-Reynoso

**Affiliations:** 1Chemical Engineering Department, University Center for Exact and Engineering Sciences, University of Guadalajara, Blvd. M. García Barragán # 1451, Guadalajara C.P. 44430, Jalisco, Mexico; javier.dnungaray@alumnos.udg.mx; 2Pharmacobiology Department, University Center for Exact and Engineering Sciences, University of Guadalajara, Blvd. M. García Barragán # 1451, Guadalajara C.P. 44430, Jalisco, Mexico; david.grajeda@alumnos.udg.mx (D.G.-A.); eire.rdelgado@academicos.udg.mx (E.R.-D.)

**Keywords:** plastic biodegradation, nitrile gloves, NBR, *Pseudomonas aeruginosa*, biofilm, EPS matrix, SEM, COVID-19, plastic pollution

## Abstract

Nitrile gloves have become a significant environmental pollutant after the COVID-19 pandemic due to their single-use design. This study examines the capability of *P. aeruginosa* to use nitrile gloves as its sole carbon energy source. Biodegradation was determined by *P. aeruginosa* adapting to increasing nitrile glove concentrations at 1%, 3%, and 5% (*w*/*v*). The growth kinetics of *P. aeruginosa* were evaluated, as well as the polymer weight loss. Topographic changes on the glove surfaces were examined using SEM, and FT-IR was used to evaluate the biodegradation products of the nitrile gloves. Following the establishment of a biofilm on the glove surface, the nitrile toxicity was minimized via biodegradation. The result of the average weight loss of nitrile gloves was 2.25%. FT-IR analysis revealed the presence of aldehydes and aliphatic amines associated with biodegradation. SEM showed *P. aeruginosa* immersed in the EPS matrix, causing the formation of cracks, scales, protrusions, and the presence of semi-spherical particles. We conclude that *P. aeruginosa* has the capability to use nitrile gloves as its sole carbon source, even up to 5%, through biofilm formation, demonstrating the potential of *P. aeruginosa* for the degradation of nitrile gloves.

## 1. Introduction

The global production and consumption of nitrile gloves has increased to satisfy industrial and healthcare protection needs. During the COVID-19 pandemic alone, usage was estimated to exceed 76 million gloves per month. Projections to 2027 indicate a sustained increase, with the demand for nitrile gloves expected to grow at a compound annual rate of between 10.6% and 11.2%. However, their single-use nature, resulting in thousands of tons being discarded, has led to a significant global issue of plastic pollution and environmental damage. Compounded by their extremely low biodegradation, these gloves can persist indefinitely in nature after being discarded. Moreover, the improper disposal of nitrile gloves has resulted in their presence in landfills, further exacerbating the contamination of marine, soil, and atmospheric environments with microplastics [1,2,3,4,5,6,7,8].

Nitrile gloves are a copolymer of acrylonitrile and butadiene (NBR: nitrile butadiene rubber). NBR is recognized as an environmental pollutant that causes soil contamination through terrestrial acidification [9,10]. Additionally, nitrile groups are known to be highly toxic, mutagenic, and carcinogenic, as reported in recent studies by Gong et al. [11] and Nigam et al. [12]. Moreover, plastics and microplastics have the potential to transport chemical pollutants, as well as microorganisms capable of forming biofilms [13].

The current literature lacks sufficient information concerning the degradability of NBR gloves and the ability of microorganisms to utilize them as a sole carbon source. Previous research studies have predominantly focused on rubber latex gloves, with limited attention given to nitrile gloves, as existing studies have focused on the monomer constituents of polymers [14,15,16,17,18,19,20].

Plastic-degrading microorganisms have the capability to use different complex polymers as sole carbon sources, and the efficiency and specificity of their degradation abilities vary a lot between species. *Pseudomonas* spp. has gained significant interest and traction in plastic biodegradation to convert, modify, and utilize toxic pollutants to convert them to harmless products in the process [21,22]. *Pseudomonas aeruginosa* is a monoflagellated, gram-negative rod, non-spore-forming bacterium. It is an obligate aerobe; however, under conditions when O_2_ is limited, it can use nitrate and arginine as a final electron acceptor. *P. aeruginosa* is a ubiquitous free-living species, commonly found in soil and water, as well as an opportunistic human pathogen; additionally, it has the ability to survive under a wide range of environmental conditions [23,24]. This bacterium possesses essential mechanisms that improve the biodegradation of synthetic plastics, like nitrile gloves; these factors are biofilm formation, enzymatic degradation, and intracellular metabolic capacities [25].

Biofilm is a highly structured microbial aggregate encased within extracellular polymeric substances (EPSs) [26]. The adhesion of biofilms to a polymer surface is mediated by physicochemical interactions, including the addition of hydrophilic functional groups, allowing them to use polymers as a substrate [27,28]. Enzymatic degradation involves extracellular and intracellular enzymes, such as depolymerases, hydrolases, hydroxylases, rhamnolipids, and proteases. These enzymes provide functional groups, such as alcohols or carbonyls, and catabolize polymer-generating monomers, reducing their molecular weight. Nitrilase is the most reported degradation enzyme in *P. aeruginosa* for highly toxic nitriles in the environment [11,14,29]. These products can be incorporated into intracellular metabolism through β-oxidation and the tricarboxylic acid cycle, ultimately being mineralized as CO_2_ or biosynthesizing products through different metabolic pathways [25,30].

*P. aeruginosa* is one of the widely reported microorganisms for the polymer degradation of low-density polyethylene, polyurethane, and polyethylene glycols via biofilm degradation, as well as polyvinyl chloride, polyvinyl alcohol, polyethylene, polystyrene, and polyphenylene sulfide [31,32]; also, *P. aeruginosa* has the capability to utilize natural rubber as a unique source of carbon and energy [33]. Nawong et al. [20] conducted a study investigating the weight loss of rubber latex gloves (0.6%, *w*/*v*) over a one-month period using individual cultures, revealing weight losses ranging from 0.5% to 9%. Badoei-Dalfard et al. [14] conducted research utilizing nitrilase from *P. aeruginosa* RZ44 to biotransform acrylonitrile at a concentration of 5 mM. Shartooh et al. [15] studied the biodegradation ability of *P. aeruginosa* utilizing acrylonitrile, and their findings revealed that the optimum concentration for biodegradation was 500 ppm, with the best adaptation period being 7 days. Egelkamp et al. [16] demonstrated the impact of nitriles on bacterial communities and their toxicity.

Despite the growing interest in biodegradation plastics, the biodegradability of nitrile gloves has not been studied in detail. Polymer biodegradation methods have not been standardized due to their complex and multi-stage nature [34]. Nevertheless, biodegradation studies apply a combination of analytical techniques to ensure accurate results, including Scanning Electron Microscopy (SEM), Fourier Transform Infrared (FT-IR) spectroscopy, bacterial growth kinetics, and weight loss measurements [35,36,37,38].

Therefore, this paper aims to address this research gap by investigating the biodegradation of nitrile gloves up to a concentration of 5% (*w*/*v*) as a sole carbon source of *P. aeruginosa* in liquid culture. The adaptation of *P. aeruginosa* with increasing concentrations of nitrile gloves, the growth kinetics of the bacteria, the presence of biofilm on the surface, the weight loss of polymers, SEM, and FT-IR spectroscopy were used to detect the biodegradation of the nitrile gloves.

## 2. Materials and Methods

### 2.1. Selection of Nitrile Glove Samples

The nitrile gloves (Ambiderm^®^ Nitrilo Soft, powder-free, medium-sized), which were commercially sourced, were cut into pieces to increase the substrate’s contact surface with the bacteria. These glove pieces were then washed with absolute ethanol for 30 min, dried in an oven at 60 °C until obtaining a constant weight, and subsequently irradiated with UV light for 5 min.

### 2.2. P. aeruginosa Cultivation

The *P. aeruginosa* strain was obtained, recovered, and purified from a plastic degradation project. For initial growth, *P. aeruginosa* was rehydrated and inoculated onto nutrient agar plates; medium contained the following constituents (gL^−1^): pancreatic digest of gelatin (5.0), beef extract (3.0), and agar (15.0). Colonies from these plates were reinoculated for selective isolation and identification of *P. aeruginosa* on cetrimide agar plates, which consisted of the following (gL^−1^): gelatin peptone (20.0), MgCl_2_ 1.4, K_2_SO_4_ (10.0), cetrimide (0.3), and agar (13.6). Subsequently, colonies from the cetrimide agar plates were transferred into Falcon tubes containing Luria Bertani Broth (LB Broth), a medium composed of the following (gL^−1^): peptone (10.0), yeast extract (5.0), and NaCl (5.0). The inoculum was then incubated at 25 °C for 48 h at 100 rpm in an orbital shaker (IKA^®^ KS 4000 i control, IKA Works, Inc., Wilmington, USA) to obtain biomass.

### 2.3. Adaptation of P. aeruginosa with 1% (w/v) Nitrile Gloves

Biomass obtained was utilized in the adaptation process with 1% (*w*/*v*) nitrile glove pieces (1 g of nitrile gloves in 100 mL of LB broth) in a shake-flask bioreactor to test microbial growing activity for degrading nitrile gloves. Shake-flask bioreactor was incubated at 25 °C for 9 days at 100 rpm in an orbital shaker. Adaptation process was performed in triplicate.

### 2.4. Biodegradation of Nitrile Gloves at 3% and 5% (w/v)

This bioprocess was performed in two phases after adaptation of *P. aeruginosa* with 1% nitrile gloves. The first phase was biodegradation with 3%, followed by the second phase with 5% nitrile glove pieces in a shake-flask bioreactor. The concentrations of 1% and 3% were selected to evaluate the enhanced microbial metabolic ability to degrade the polymer, with escalating concentrations of nitrile gloves leading to an improved adaptation to the new substrate. The primary focus of our biodegradation study was on the highest concentration of nitrile gloves (5%).

The shake-flask bioreactor with glove pieces at 3% (*w*/*v*) (8.25 g of nitrile gloves in 275 mL of LB broth) was inoculated with 5 mL of biomass of *P. aeruginosa* from the adaptation process at 1%. The shake-flask bioreactor was incubated at 25 °C for 14 days at 100 rpm in an orbital shaker.

Biodegradation of glove pieces at 5% (*w*/*v*) (13.75 g of nitrile gloves in 275 mL of LB broth) was tested with 5 mL of biomass of *P. aeruginosa* from the shake-flask bioreactor for nitrile gloves at 3%. The shake-flask bioreactor was incubated at 25 °C for 7 days at 100 rpm in an orbital shaker.

These biodegradation processes were performed in triplicate. Also, we included two control experiments. The first control consisted of a shake-flask bioreactor containing only LB broth as abiotic control, while the second control contained nitrile glove pieces in LB broth to ensure that any degradation of polymer was caused by *P. aeruginosa* and not by the medium or environmental factors.

### 2.5. Growth Kinetics of P. aeruginosa Using Nitrile Gloves as Sole Carbon Source

Microbial growth was determined by measuring the colony-forming units (CFUs) in CFUmL^−1^ (the counts were natural logarithm (ln)-transformed) by a microdilution plating method that was carried out from the shake flasks of biodegradation with 5% nitrile gloves every 24 h for 7 days, taking 2 mL into vials and serially diluting it 1:10 in tubes containing LB broth in a range of 1 × 10^−1^ mL to 1 × 10^−6^ mL and inoculating it onto Mueller–Hinton agar plates; the medium contained the following constituents (gL^−1^): beef extract powder (2.0), acid digest of casein (17.5), starch (1.5), and agar (17.0). Plates with colony counts within the range of 25 to 250 were selected to calculate the doubling time and assess how fast *P. aeruginosa* can grow under the biodegradation condition of 5% nitrile glove pieces. Doubling time was calculated using Equation (1) [39]:(1)$$td=\frac{\ln2}{\mu_{max}}$$
where *td* is doubling time expressed in days, ln is natural logarithm, and μ*_max_* is the maximum specific growth rate with days^−1^ as the units.

### 2.6. Optical Density (OD 600 nm) Data for Biodegradation of Nitrile Gloves

Biodegradation of nitrile gloves was measured by monitoring changes in optical density (OD) at 600 nm using the turbidity as an indirect method to evaluate the survival and growth of *P. aeruginosa*, as well as the production of extracellular polymeric substances (EPSs) and degradation products [40,41]. The experiments were performed with the different percentages of nitrile glove pieces (1%, 3%, and 5%), and 2 mL aliquots were taken from each of the flasks with the nitrile gloves. The control, which involved nitrile glove pieces in LB broth, was utilized as a baseline for comparison. Subtracting the control’s result from the other experiments isolated the specific impact of *P. aeruginosa* on polymer degradation, removing potential influences from the medium or environmental factors.

The adaptation at 1% was determined after a 9-day incubation period, while biodegradation at 3% was determined after a 14-day incubation period. Biodegradation at 5% was monitored every 24 h for 7 days using a UV-Vis spectrophotometer (Jenway^TM^ Genova, Thermo Scientific^®^, Waltham, MA, USA). The percentage change in OD was calculated with Equation (2) [42]:(2)$$\Delta \%\;OD=\frac{OD\;final-OD\;initial}{OD\;initial}\;*\;100$$
where Δ% OD is the percentage change in optical density, and OD initial and OD final are the optical density of incubation at initial and final period, respectively.

### 2.7. Weight Loss of Nitrile Gloves

To quantify the mass reduction in nitrile gloves by *P. aeruginosa*-mediated degradation, the pieces of nitrile gloves were collected from the shake-flask bioreactor for biodegradation at 5% (g/L) after the 7-day incubation period, and the weight (g) of the nitrile gloves was measured before and after degradation. Percentage of weight loss was calculated using Equation (3) [35]:(3)$$Weight\;loss\%=\frac{W1\;-\;W2}{W1}\;*\;100$$
where W1 is the weight (g) of the nitrile gloves before degradation and W2 is the weight (g) of the nitrile gloves after degradation.

### 2.8. Fourier Transform Infrared Spectroscopy (FT-IR)

FT-IR (Nicolet^TM^ i550^TM^ FT- IR, Thermo Scientific^®^, Waltham, USA) was used to analyze chemical structural changes in nitrile gloves following biodegradation by *P. aeruginosa*. The absorption bands of the unknown components were correlated with the known absorption frequencies. Aliquots of 2 mL were taken from each of the shake flasks of nitrile gloves (adaptation at 1%, biodegradation at 3% and 5%) at the beginning and the end of each bioprocess.

The collected aliquots were dried at 60 °C for 12 h until desiccation, after which their FT-IR spectra were obtained using attenuated total reflectance with a 4400–400 cm^−1^ range diamond tip.

### 2.9. Microscopic Analysis of Surface Changes in Nitrile Gloves

To observe surface modifications in nitrile gloves, samples were taken from shake flasks at the beginning and the end of each bioprocess (adaptation at 1%, biodegradation at 3% and 5%). These samples were observed using the Motic^®^ BA310 LED Digital, with 10× objective.

### 2.10. Scanning Electron Microscopy (SEM)

SEM was conducted to analyze the physical changes that caused degradation on plastic surfaces [20]. Furthermore, to examine the attachment of *P. aeruginosa* to nitrile gloves, samples were dried at 60 °C to remove moisture and immediately imaged using SEM. Samples were taken from adaptation at 1% and biodegradation with 3% and 5% nitrile gloves and two controls: nitrile gloves without any treatment and the abiotic control.

The pieces of the nitrile gloves that did not exceed the diameter size of the pin were selected and were mounted onto pins of the Scanning Electron Microscope (FE-SEM Microscope, MIRA^TM^ 3LMU, TESCAN a.s., Warrendale, Pittsburgh, PA, USA) using an electrically conductive carbon tape and coated with a layer of gold (for 40 s, at 15 kV, at 301 μA, with air as sputter gas, with magnifications of 1.5 k×, 8.5 k×, and 15 k×), producing a 12.64 nm-thick layer [43].

## 3. Results

### 3.1. Growth Kinetics of P. aeruginosa Using Nitrile Gloves as Sole Carbon Source

*P. aeruginosa* showed growth kinetics similar to those of a typical pure culture upon utilizing nitrile gloves as its sole carbon source; data collected for the growth curve of *P. aeruginosa* in biodegradation at 5% are shown in Figure 1, where it was found that the highest increase in biomass occurred on the second day, being the peak in the growth of the bacteria. From this day, the CFUmL^−1^ count of *P. aeruginosa* began to decrease, reaching 2 ln lower than the amount on day 0 of biodegradation. Therefore, even at a high concentration of the polymer, with 5% nitrile gloves, *P. aeruginosa* significantly increased its biomass, indicating its ability to use nitrile gloves as its sole carbon source.

A maximum growth rate (μ_*max*_) of 2.0374 days^−1^ was obtained, with a doubling time (td) of 0.3401 days, which is equivalent to 8.16 h. Therefore, considering that the log phase lasted 2 days, bacterial duplication occurred 5.88 times, as calculated using Equation (1).

### 3.2. Optical Density (OD 600 nm) Data for Biodegradation of Nitrile Gloves

Changes in OD were observed with different percentages of nitrile gloves as the sole carbon source, which showed variations in bacterial density and the presence of metabolic and degradation products, as shown in Figure 2. Applying Equation (2), the Δ% OD in the adaptation phase was 87%, while the Δ% OD during biodegradation with 3% nitrile gloves was 74%.

These values indicate the ability of *P. aeruginosa* to grow in the presence of polymer as its sole carbon source and its tolerance to increasing percentages of nitrile gloves. This can be explained by how *P. aeruginosa* utilizes mechanisms to use nitrile molecules as an energy source and adapts during lag phase. During the biodegradation phase at 3%, *P. aeruginosa* faced an increased concentration of nitrile gloves, which could lead to starvation over the 14-day biodegradation period. Initially, this increased concentration could be toxic until *P. aeruginosa* developed optimal metabolic pathways to degrading the nitrile gloves in the subsequent nitrile concentration of 5% [16]. 

During biodegradation at 5%, the change in OD of 600 nm over 7 days revealed an increase in bacterial density resulting from viable and dead bacteria, metabolic waste products, extracellular polymeric substances, or degradation products of the nitrile gloves. The data obtained showed a 30% increase in OD from the second day to the third day, which is similar to the CFUmL^−1^ results. The OD changes during biodegradation at 5% are shown in Figure 3.

The OD increase demonstrates the ability of *P. aeruginosa* to survive and grow using nitrile gloves at 5% as its sole carbon source. This was observed through changes in color, increased turbidity, thickness, and the formation of biofilm on the glove surfaces. During this phase, the Δ% OD was 153%, as calculated using Equation (2). The previous result can be attributed to heightened microbial metabolic capacity for degrading the polymer with increasing concentrations of nitrile gloves, ultimately resulting in an enhanced adaptation to the new substrate. 

### 3.3. Weight Loss of Nitrile Gloves

During the biodegradation phase at 5% (13.75 g of nitrile gloves), *P. aeruginosa* demonstrated its capability to cause mass reduction in nitrile gloves. The resulting percentage of weight loss of nitrile gloves was 2.25% (0.3071 g), with an SD of ±0.81% (0.1098 g). In comparison, the control samples showed a weight loss of 0.59% (0.0796 g), with an SD of ±0.04% (0.0048 g). These results are shown in Figure 4.

The 2.25% reduction in the weight of the nitrile gloves is significant because according to this percentage, complete degradation of the pieces of nitrile gloves would occur within 311.1 days (<1 year), contrasting with the 35.4-year (3.1 × 10^5^ h) lifetime of NBR (0.5 g) at 50 °C that was based on the measurements of thermal biodegradation taken by Kawashima and Ogawa [44]. Our results are consistent with those of Nawong et al. [20], who reported weight losses ranging from 0.5% to 9% in rubber latex gloves (0.6%, *w*/*v*), with a 0.2% loss in their abiotic control after 30 days of exposure to unidentified cultures sourced from soil samples. This suggests potential interactions between the medium and the polymer.

These findings highlight the potential of our approach to biodegrade nitrile gloves using *P. aeruginosa* by enabling its survival under the specified conditions and using this polymer as its sole carbon energy source, ultimately resulting in a reduction in the weight of nitrile gloves.

### 3.4. Fourier Transform Infrared Spectroscopy (FT-IR)

FT-IR data revealed compositional changes in the LB medium following the biodegradation of nitrile gloves by *P. aeruginosa* over a 7-day period. Four vibrational bands with major importance were identified in the FT-IR spectra of the control samples, corresponding to alkanes at 2956 cm^−1^ (C-H stretch) and 1453 cm^−1^ (C-H bend), primary amines (N-H) at 1580 cm^−1^, and aliphatic amines (C-N) at 1080 cm^−1^ [45,46,47]. These functional groups are characteristic of organic molecules found in LB, such as peptones. Table 1 shows the FT-IR peaks present after the biodegradation process of nitrile gloves by *P. aeruginosa* in LB.

The FT-IR spectrum of the LB control (nitrile glove pieces in LB broth) shows consistent responses and the presence of the same functional groups, indicating that the composition of the LB remained unchanged over the period of the biodegradation process; only changes in transmittance were observed. Spectra of control samples from the initial and final stages are shown in Figure 5a. FT-IR spectra of the different phases of biodegradation (adaptation at 1%, biodegradation at 3%, and biodegradation at 5%) showed the metabolic capability of *P. aeruginosa* to utilize nitrile gloves as a carbon energy source, as shown in Figure 5b.

The FT-IR results of the biodegradation phases showed two new vibrational bands that emerged at 1738 cm^−1^, corresponding to aldehydes (C=O), and a second peak of aliphatic amines (C-N) at 1216 cm^−1^. The presence of C=O suggests the conversion of unsaturated carbon bonds and the decreased presence of the nitrile group in nitrile gloves [48]. Similar results were reported by Nawong et al. [20] in their study on rubber latex gloves, where they observed the presence of aldehydes close to the analyzed FT-IR region (at 1634 cm^−1^). These new vibrational bands of aldehydes and aliphatic amines are product characteristics of bacteria metabolism that could be identified as a product of biodegradation of the polymer of nitrile gloves; this is evidence of the ability of *P. aeruginosa* to grow utilizing carbon chains derived from synthetic polymers such as nitrile [18,33,49].

### 3.5. Microscopic Analysis of Surface Changes on Nitrile Gloves

Changes were observed after the adaptation of *P. aeruginosa* with 1% nitrile gloves, showing a loss of homogeneity in texture (integrity) and the presence of protrusions or roughness, as shown in Figure 6a. Comparing samples of nitrile gloves between the processes of adaptation at 1% and biodegradation at 3%, visual development of a biofilm produced by *P. aeruginosa* was observed at the periphery of the nitrile glove, as shown in Figure 6b (indicated by arrow), suggesting an increase in the expression of the genes necessary for the production of EPSs, which allowed the strain to attach to the surface of the plastic and to use and modify it as a substrate [27,28]. The surface structure of the nitrile glove was completely modified in biodegradation at 5%, and as shown by the arrow in Figure 6c, we can observe a surface with irregularities throughout the nitrile gloves, where convex bubble-shaped areas were generated due to the interaction of *P. aeruginosa* with the polymeric surface. We also detected irregularities in the integrity of the plastic, and the surface showed fissures, demonstrating the deterioration caused by the bacteria.

### 3.6. Scanning Electron Microscopy (SEM)

*P. aeruginosa* was observed attached to the surfaces of pieces of nitrile gloves at each phase of biodegradation. The surfaces of nitrile gloves during adaptation at 1% appeared rough, with irregular reliefs and craters, and the EPS matrix of *P. aeruginosa* was observed covering the substrate and providing strong adhesion to the bacteria in the lightest areas, as shown in Figure 7a–c, and surface irregularities such as cracks, scales, and lobes were observed on the surface.

The surface relief of nitrile gloves during biodegradation at 3% exhibited more irregular protrusions and a greater number of concave spaces, and scattered rod-shaped cells can be seen immersed in the EPS matrix on the surface of the nitrile glove, with many cracks and scales, in Figure 7b. Biodegradation at 5%, as shown in Figure 7c, exhibited the most pronounced relief in the form of the granules and the largest scattered crater.

Additionally, rods were rarely observed individually or forming chains, indicating the presence of *P. aeruginosa* immersed in the EPS matrix on the surface of the gloves, as depicted in Figure 8a. The produced craters were extensive, and the area where bacterial adhesion to the surface was most prominent was identified. Semi-spherical particles were observed on the same surface, possibly being remnants of the nitrile glove piece. The reliefs appeared amorphous, and the semi-spherical particles resulting from the glove biodegradation were more clearly visible [50]. The nitrile gloves without treatment exhibited slight irregularities due to their elastic properties, revealing the topographic characteristics of the nitrile gloves. As the magnification increased, the surface appeared smoother, as illustrated in Figure 7d and Figure 8b.

*P. aeruginosa*, its adhesion to the surface with the EPS matrix of the biofilm, and the effects resulting from the biodegradation process, including the depolymerization of the nitrile gloves, provide additional support for the FT-IR findings. These processes led to significant topographic changes on the surface, characterized by the formation of craters where the bacteria concentrated and carried out their metabolic activities, which was consistent with surface morphology findings from other polymers [20,34,35,50,51].

## 4. Discussion

The ability of *P. aeruginosa* to adapt and survive without additional carbon sources generated the biodegradation of nitrile gloves, with biofilm formation being a crucial factor in adapting to adverse conditions. *P. aeruginosa* demonstrated a capability to grow with 1% nitrile gloves as its sole carbon source and showed its tolerance to utilize this polymer. However, at a higher concentration of 3%, a minor change in OD was observed due to nitrile toxicity, and the result was starvation of *P. aeruginosa* over the 14-day biodegradation period [16]. Remarkably, under increasing concentrations of nitrile gloves, this condition developed an optimal metabolic capacity for degrading the gloves. This was demonstrated through the ability to grow with 5% nitrile gloves, leading to significant biomass production. These findings underscore the importance of preadapting the bacterial strain to biodegrade polymers by gradually increasing the polymer concentration until significant results are achieved, even under unfavorable conditions.

Additionally, a weight loss of 2.25% was obtained in the nitrile gloves over a 7-day biodegradation period, indicating the capability of *P. aeruginosa* to assimilate them as a carbon source and decrease the lifetime of nitrile gloves. The FT-IR analysis revealed characteristic changes associated with biodegradation, including the presence of aldehydes and aliphatic amines, as a result of the capability of *P. aeruginosa* to utilize nitrile gloves as a carbon source. As the percentage of nitrile gloves increased, the spectral resolution improved, indicating the adaptation and assimilation of nitrile molecules by *P. aeruginosa* during biodegradation.

Biofilm was observed on the periphery of the nitrile gloves through microscopic analysis, which is consistent with the OD results at 3% concentration. These findings suggest the development of mechanisms by *P. aeruginosa* to utilize nitrile gloves as a sole carbon source, involving the expression of genes required for the production of an EPS matrix for attachment, including exopolysaccharides, proteins, lipids, and extracellular DNA (eDNA) [52,53]. To our knowledge, we are the first group to demonstrate the use of *P. aeruginosa* in degrading nitrile gloves, thereby evidencing its biofilm properties that facilitate nitrile biodegradation.

On the other hand, *P. aeruginosa* demonstrated a capability to modify the surface topography of the nitrile glove, resulting in the formation of convex lobe-shaped structures, as well as the presence of cracks and scales. *P. aeruginosa* could be observed immersed in the EPS matrix on the surface of the nitrile gloves through SEM. Similar observations were made in relation to the percentage of nitrile gloves used. *P. aeruginosa* exhibited its ability to modify the surface, which was related with its adaptability to utilize nitrile gloves as its sole carbon source when the percentage was increased. This was evident in the presence of the extensive craters observed on the surface of the gloves, where the biofilm of *P. aeruginosa* was identified, as well as the protrusions and depolymerization indicated by the presence of semi-spherical holes and particles.

Despite the nitrile glove industry benefiting from advantageous NBR properties, such as hydrophobicity and cytotoxicity towards microorganisms due to the acrylonitrile that minimizes bacterial growth on the surface of nitrile gloves, Akbarian-Saravi et al. [54] concluded in their research that NBR is nonbiodegradable under aerobic conditions. However, their study utilized an aerobic inoculum from a wastewater treatment plant and did not account for the biofilm properties of bacteria. Our findings indicate that *P. aeruginosa* can attach to the surface of nitrile gloves through its biofilm mechanism, facilitating polymer biodegradation and its abilities for both aerobic and anaerobic respiration, allowing *P. aeruginosa* to switch between aerobic oxidation and denitrification under anaerobic conditions [55,56,57]. This highlights two important implications: firstly, for the nitrile glove industry, our results demonstrate the feasibility of employing *P. aeruginosa* biofilms in developing biodegradation processes to mitigate plastic pollution. Secondly, for healthcare centers and hospitals, there is a potential risk of contamination from *P. aeruginosa* biofilms on nitrile gloves.

## 5. Conclusions

This study concludes that *Pseudomonas aeruginosa* has the capability to use nitrile gloves as a sole carbon source and degrade them at a concentration of up to 5% (*w*/*v*) by biofilm formation on the surface. Furthermore, the increase in the concentration of nitrile gloves leads to the development of mechanisms such as the production of an EPS matrix, which helps *P. aeruginosa* tolerate nitrile toxicity and establish optimal metabolic pathways for polymer degradation. These findings demonstrate the potential of *P. aeruginosa* to be utilized in the degradation of nitrile gloves, which have recently emerged as a notable environmental pollutant. This insight may serve as a reference for developing more-effective plastic treatment strategies.

## Figures and Tables

**Figure 1 polymers-16-01162-f001:**
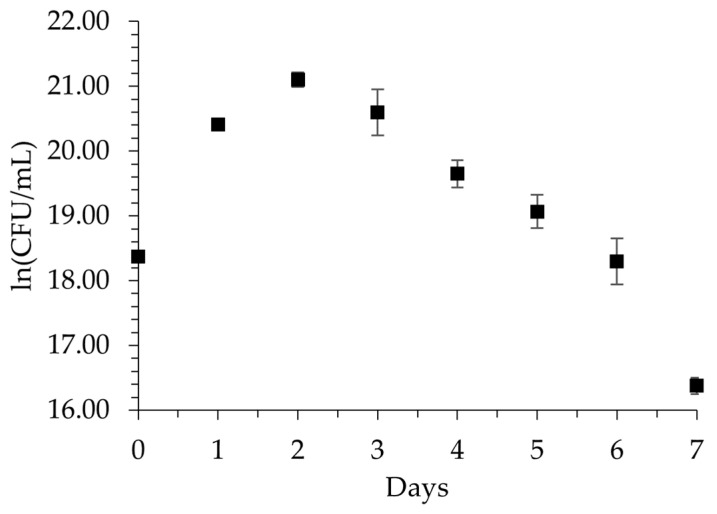
Growth changes in ln (CFUmL^−1^) of *P. aeruginosa* during biodegradation with 5% nitrile gloves in Luria broth over 7 days. Data are represented as mean values ± SD (n = 3).

**Figure 2 polymers-16-01162-f002:**
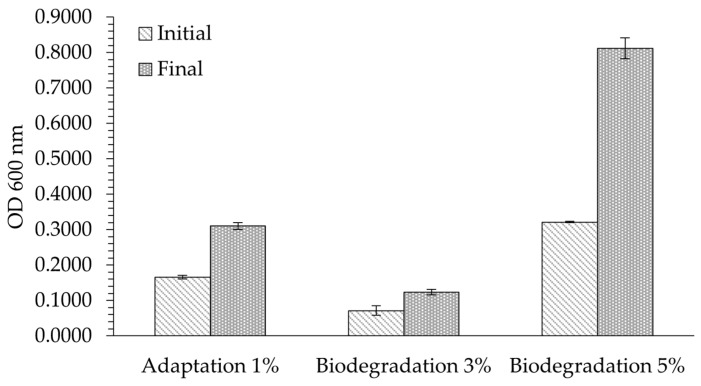
Comparison of initial and final OD values with different percentages of nitrile gloves as the sole carbon source: adaptation of 1% after 9 days, biodegradation of 3% after 14 days, and biodegradation of 5% after 7 days. Results are presented as mean values ± SD (n = 3). The control was utilized as a baseline for comparison.

**Figure 3 polymers-16-01162-f003:**
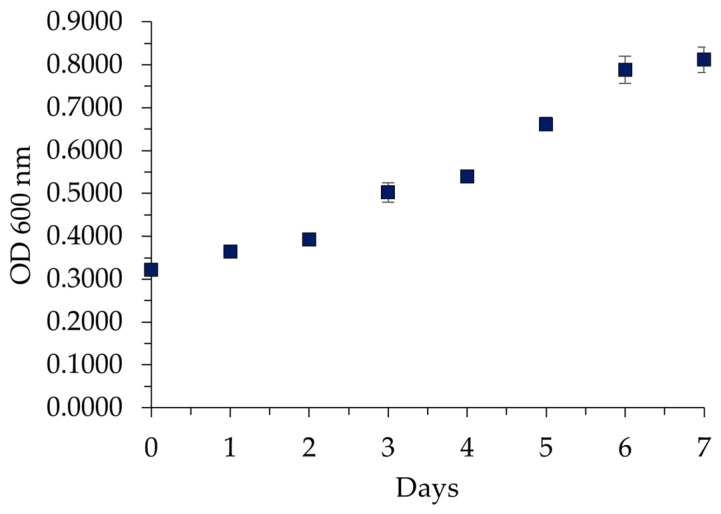
Optical density changes in *P. aeruginosa* during biodegradation with 5% nitrile gloves in Luria broth over 7 days. Data represented as mean values ± SD (n = 3). The control served as the baseline for comparison.

**Figure 4 polymers-16-01162-f004:**
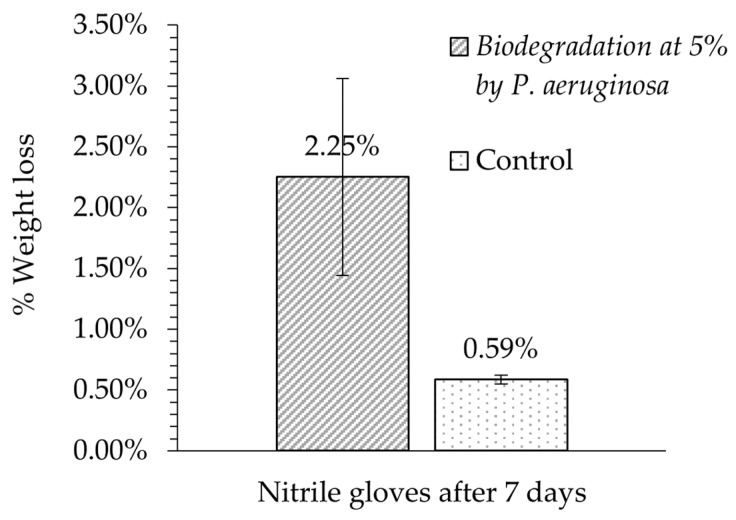
Weight loss of nitrile gloves caused by *P. aeruginosa* during biodegradation at 5% after 7 days. The mean % weight loss ± SD of the nitrile gloves was 2.25% ± 0.81%, and for the control samples, it was 0.59% ± 0.04% (n = 3).

**Figure 5 polymers-16-01162-f005:**
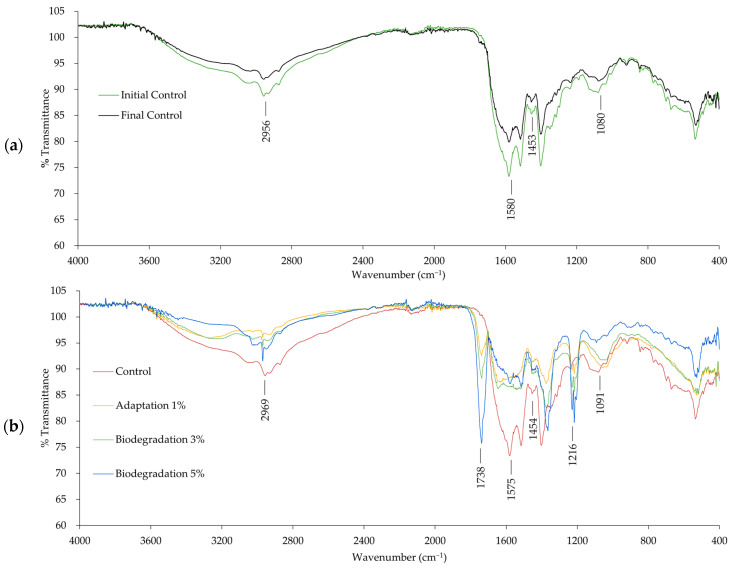
The FT-IR spectra of the nitrile gloves degraded by *P. aeruginosa* with different percentages of polymer in LB: (**a**) FT-IR spectrum of control at initial stage and final stage after 7 days; (**b**) FT-IR spectrum of degraded nitrile gloves with adaptation at 1% after 9 days, biodegradation at 3% after 14 days, and biodegradation at 5% after 7 days.

**Figure 6 polymers-16-01162-f006:**
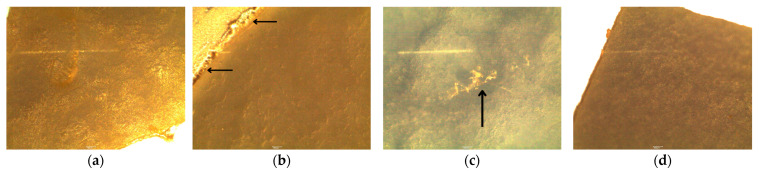
Surface changes caused by *P. aeruginosa* on nitrile gloves in different percentages (*w*/*v*) at the end of each bioprocess: (**a**) adaptation of *P. aeruginosa* at 1%; (**b**) biodegradation at 3%; (**c**) biodegradation at 5%; (**d**) nitrile glove control (no bacteria). Nitrile gloves were observed by Motic^®^ BA310 LED Digital, 10× magnification.

**Figure 7 polymers-16-01162-f007:**
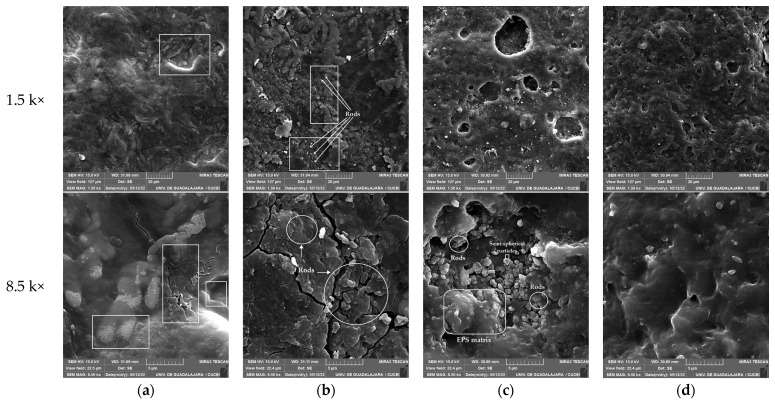
Biodegradation of nitrile gloves by *P. aeruginosa* viewed with SEM: (**a**) adaptation at 1%; (**b**) biodegradation at 3%; (**c**) biodegradation at 5%; (**d**) control (nitrile gloves without treatment). Magnification 1.5 k× and 8.5 k×.

**Figure 8 polymers-16-01162-f008:**
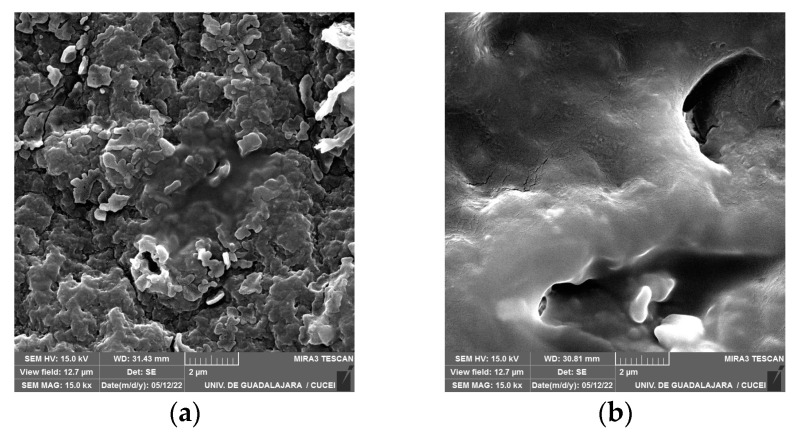
Biofilm of *P. aeruginosa* on surface of nitrile gloves viewed with SEM at 15 k×: (**a**) biodegradation at 5% and (**b**) control (nitrile gloves without treatment).

**Table 1 polymers-16-01162-t001:** Functional groups after biodegradation of nitrile gloves by *P. aeruginosa* in LB.

FT-IR Peak (cm^−1^)	Region (cm^−1^)	Functional Group	References
2969	3000–2840	Alkane (C-H stretch)	[18,33,45,46,47]
1575	1560–1640	Primary amine (N-H)	
1454	1475–1450	Alkane (C-H bend)	
1092	1020–1220	Aliphatic amines (C-N)	
1738	1720–1740	Aldehydes (C=O)	
1216	1020–1220	Aliphatic amines (C-N)	

## Data Availability

Data are contained within the article.

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
