# Peer review of "Biodegradation of Nitrile Gloves as Sole Carbon Source of Pseudomonas aeruginosa in Liquid Culture"

_polymers, 2024, doi:10.3390/polym16081162_

Round 1

Reviewer 1 Report

Comments and Suggestions for Authors

The authors of this study provide a very useful and relevant investigation into the biodegradation of medical nitrile rubber gloves which have been and are still heavily utilised. Essentially the outcome is to seek an effective bacterium that degrades the material so that it may be effectively composted. The data appears reliable and undertaken in the usual professional manner and is clearly presented for the lay reader. The discussion integrates the literature in  a critical and professional format with what appears to be a very useful outcome. 

Author Response

Reviewer #1:

  • The authors of this study provide a very useful and relevant investigation into the biodegradation of medical nitrile rubber gloves which have been and are still heavily utilised. Essentially the outcome is to seek an effective bacterium that degrades the material so that it may be effectively composted. The data appears reliable and undertaken in the usual professional manner and is clearly presented for the lay reader. The discussion integrates the literature in  a critical and professional format with what appears to be a very useful outcome.

We are grateful to the reviewer # 1 for the positive evaluation of our manuscript. We appreciate acknowledgement of the critical integration of literature into the discussion. Furthermore, thank you for recognizing the usefulness and relevance of our research in the field of biodegradation. Your feedback motivates us to continue our research efforts in this important area of study.

- In the italic text : the reviewer’s comments.

In bold notation: authors’ responses

Reviewer 2 Report

Comments and Suggestions for Authors

How was biodegradation procedure chosen? Did  you took someone elses research or it was done according to standard procedures?

The  figures FTIR spectra needs to be improved -  the colors of lines does not allow to see which color corresponds to which test (looks the same). In addition, the transmitance y-axes needs to be on the left side of the graph (crossing at the higest number). The FTIR spectra is composed of vibrational band not of signals and peaks (its a sleng).

The equations in the manuscript should be written in required chemical form with appropriate fractions, etc. not written in one line. In addition, use the standard form of equations writing and symbols explanation (below equation).

The Refrence list is not prepaired according to Journal requirements.

Round 2

Reviewer 2 Report

Comments and Suggestions for Authors

Authors made all changes needed.